# Domain Decomposition Spectral Method Applied to Modal Method: Direct and Inverse Spectral Transforms

**DOI:** 10.3390/s22218131

**Published:** 2022-10-24

**Authors:** Kofi Edee, Gérard Granet, Francoise Paladian, Pierre Bonnet, Ghida Al Achkar, Lana Damaj, Jean-Pierre Plumey, Maria Cristina Larciprete, Brahim Guizal

**Affiliations:** 1INP, CNRS, Institut Pascal, Université Clermont Auvergne, F-63000 Clermont-Ferrand, France; 2Kavli Institute of Theoretical Physics (KITP), University of California, Santa Barbara, CA 93106, USA; 3Dipartimento di Scienze di Base ed Applicate per l’Ingegneria, Sapienza Universita di Roma, Via A. Scarpa 16, I-00161 Roma, Italy; 4Laboratoire Charles Coulomb (L2C), UMR 5221 CNRS-Université de Montpellier, F-34095 Montpellier, France

**Keywords:** metasurfaces, metalens

## Abstract

We introduce a Domain Decomposition Spectral Method (DDSM) as a solution for Maxwell’s equations in the frequency domain. It will be illustrated in the framework of the Aperiodic Fourier Modal Method (AFMM). This method may be applied to compute the electromagnetic field diffracted by a large-scale surface under any kind of incident excitation. In the proposed approach, a large-size surface is decomposed into square sub-cells, and a projector, linking the set of eigenvectors of the large-scale problem to those of the small-size sub-cells, is defined. This projector allows one to associate univocally the spectrum of any electromagnetic field of a problem stated on the large-size domain with its footprint on the small-scale problem eigenfunctions. This approach is suitable for parallel computing, since the spectrum of the electromagnetic field is computed on each sub-cell independently from the others. In order to demonstrate the method’s ability, to simulate both near and far fields of a full three-dimensional (3D) structure, we apply it to design large area diffractive metalenses with a conventional personal computer.

## 1. Introduction

Regardless of the numerical method used to solve an electromagnetic (EM) problem, the number of unknowns of the field components increases as well as the electrical size of the device. Consequently, the modeling of extended structures can be time and memory consuming for a conventional computer. This makes the numerical simulation of large-scale problems more challenging for numerical simulations, and it is still an open task. Of course, in practice, the meaning of fast or large depends on the performance of the employed computer resources.

In order to overcome this challenging task and to ensure fast and reliable numerical solutions, two main points could be considered. The first one is the way the code is implemented, and the second one is focused on the numerical method used to solve equations at hand.

For a given electromagnetic problem stated on a large-scale domain, one of the first intuitive methods to overcome the large size issues is the sub-structuring of its computational domain. This technique consists of dividing the whole domain into a sequence of sub-domains. The solution inside these sub-domains may be found using an iterative or parallel strategy. Concerning the iterative process, the solution of the equation is found after solving the boundary conditions equations. This approach is considered within the context of the iterative domain decomposition techniques such as Schwarz’s method [1,2,3,4,5,6]. However, when the number of interfaces increases, the strategy can be time consuming. On the other hand, the parallel computing strategy relies on overlapping [7] or stitching [8] the grid of the sub-domains without applying any boundary conditions. Concerning the numerical method, a large class of electromagnetic field simulation for photonics applications requires solving the partial differential equations (PDE) obtained from Maxwell’s equations and a set of ad hoc boundary conditions. A priori, there is no unique and universal method to efficiently solve the PDEs obtained from Maxwell’s equations in the general case. Among these methods, some clearly appear to be more used than others, either for their simplicity of implementation, or for their ability to take into account complex-geometry problems, or because they are the most efficient in terms of numerical accuracy. Yet, finite difference methods [9,10], finite element methods [11,12], or spectral modal methods [13,14,15,16,17,18,19,20,21] are considered from these methods. Furthermore, all these numerical methods involve different amounts of theoretical effort which yield different computational efficiencies. It is clear that improving the efficiency of the numerical implementation and process will often require more pre-computational analytical effort as well as more in-depth understanding and usage of electromagnetic theory. For instance, spectral modal methods aim to approximate an unknown function, as a solution of a differential or an integral equation, by a finite sum of eigenmodes. These eigenmodes are expanded into a so-called basis function. In the case of finite difference and finite elements methods, many studies have been conducted to make them suitable to large-scale problems [7,22,23].

Contrary to these methods, the spectral modal methods, in spite of many previous developments [7,8], still face some major challenges to find numerical solutions for some large-scale problems.

In this paper, we propose a mathematical formalism based on a projector enabling the projection of any eigenvector defined on a domain Ω into a sequence of eigenfunctions based on smaller-scale boundary value problems defined on subdomains ${\tilde{\Omega }}_{ij}$. In the proposed domain decomposition method, a large-scale surface is squared into sub-cells. Hence, a projector, i.e., a link between the large problem and the small one, is introduced. This projector, which is defined from the sequence of eigenfunctions of both large and small problems, univocally enables the association of the spectrum of any electromagnetic field component related to a large-size problem, with its restriction on the small-size systems. Consequently, the spectrum of the electromagnetic field component on each sub-cell may be simulated independently. This makes the proposed approach suitable to be integrated in a parallel computing scheme in order to allow for accounting larger size systems.

This paper is organized as follows: After the introduction (Section 1), we introduce in Section 2 the physical system and the proposed domain decomposition spectral method (DDSM). The DDSM is explained and described in detail, and the concept of direct and inverse transform is presented. Moreover, Section 3 is devoted to applications in the framework of the modal method. In this section, we introduce the bases of the modal method applied to a 3D problem. In a first illustration, we demonstrate its ability to accurately describe a field radiated by an electric dipole. In a second example, we prove the capability of the DDSM to simulate the near and far fields of a high-refractive-index-deeply-etched Fresnel plate zone, metalenses made of dielectric cylindrical waveguides (nanorods) with variables cross-section and a dielectric Pachatarnam–Berry phase metalens.

## 2. Methods

The framework of the proposed domain decomposition spectral method (DDSM) is the spectral modal method. For the reader that is not familiar with this method, we highlight in the following first two subsections the main underlying principle both of the modal and the spectral methods.

### 2.1. Maxwell’s Equations and Eigenvalue Problem

Let us consider Maxwell’s equations [20] in a Cartesian coordinates system $\left(O,e_{x},e_{y},e_{z}\right)$ and the constitutive relations stated on a 2D bounded domain $\Omega =\left[-0.5D_{x},0.5D_{x}\right]\times \left[-0.5D_{y},0.5D_{y}\right]]\subset R^{2}$. We assume that the physical parameters of the medium, namely the relative permittivity ε(x,y) and the permeability μ(x,y) functions, do not depend on the *z* variable. In this case, any component of the electromagnetic field can be written as $\Phi \left(x,y\right)exp\left(-ik_{0}\gamma z\right)$, where $k_{0}=\omega \sqrt{\varepsilon_{0}\mu_{0}}$ is the wavenumber in the vacuum, ω is the angular frequency and $k_{0}\gamma$ is the propagation constant along the *z* direction. $\varepsilon_{0}$ and $\mu_{0}$ are the permittivity and the permeability of the vacuum. It can be shown that the transverse components of the electromagnetic field $E_{x}$, $E_{y}$, $H_{x}$ and $H_{y}$ satisfy an eigenvalue equation:(1)$$L_{\mathrm{EH}}L_{\mathrm{HE}}|\Phi \rangle =-\gamma^{2}|\Phi \rangle$$
where $\Phi ={\left[E_{x},E_{y}\right]}^{t}$, and
(2)$$\left\{\begin{array}{c} L_{EH}=\frac{1}{k_{0}^{2}}\left[\begin{array}{cc} -\partial_{x}E^{-1}\partial_{y} & \partial_{x}E^{-1}\partial_{x}+k_{0}^{2}\mu \\ -\partial_{y}E^{-1}\partial_{y}-k_{0}^{2}\mu & \partial_{y}\partial_{x} \end{array}\right]\\ L_{HE}=\frac{1}{k_{0}^{2}}\left[\begin{array}{cc} -\partial_{x}\mu^{-1}\partial_{y} & \partial_{x}\mu^{-1}\partial_{x}+k_{0}^{2}E\\ -\partial_{y}\mu^{-1}\partial_{y}-k_{0}^{2}E & \partial_{y}\partial_{x} \end{array}\right] \end{array}\right.,$$
The eigenvalue Equation (Equation 1) is solved using the spectral method briefly described in the next subsection.

### 2.2. Spectral Method and Modal Method

In electromagnetism, a spectral method aims to describe a function |Φ〉 representing any electromagnetic field component with a set of finite expansion coefficients $\Phi^{nm}$ on a selected basis also known as expansion functions $|e_{nm}\rangle$:(3)$$|\Phi \rangle =\Phi^{nm}|e_{nm}\rangle$$
by using Einstein’s convention. Expansion or weighted coefficients $\Phi^{nm}$ are the spectrum of the expanded function Φ with respect to the sequence of expansion vectors $|e_{nm}\rangle$. In a Cartesian coordinates system with axis (O,x), (O,y), (O,z), and in the framework of the modal method, the structure under consideration is often divided into some layers denoted $I_{z}^{\left(l\right)}$ with respect to the propagation direction that is assumed to be (O,z) in our case. For example, in Figure 1, a 2D-diffraction grating is divided into 4 layers $I_{z}^{\left(1\right)}$, $I_{z}^{\left(2\right)}$, $I_{z}^{\left(3\right)}$ and $I_{z}^{\left(4\right)}$ with respect to the propagation direction.

In any of the layers $I_{z}^{\left(l\right)}$, the physical parameters, namely the permittivity and permeability, are described as functions that do not depend on the variable *z*. Consequently, each component of the electromagnetic field may be expanded on a set of eigenfunctions, i.e., solutions $|\Phi_{kl}\rangle$ of an eigenvalue equation similar to Equation (Equation 1):(4)$$L|\Phi \rangle =-\gamma^{2}|\Phi \rangle .$$
The completeness of the set of eigenvectors $(|\Phi_{kl}\rangle {)}_{kl}$ allows one to expand any function |f〉 defined on a domain Ω as a linear combination of the eigenvectors $|\Phi_{kl}\rangle$:(5)$$|f\rangle =A^{kl}|\Phi_{kl}\rangle .$$
Regarding Equation (Equation 5), the modal method can be viewed as a spectral method using as basis functions a set of eigenvectors $|\Phi_{kl}\rangle$ of the operator L. The Fourier modal method (FMM) consists of expanding the eigenvectors $|\Phi_{kl}\rangle$ in terms of a generalized Fourier basis:(6)$$|\Phi_{kl}\rangle =\Phi_{kl}^{nm}|e_{nm}\rangle ,$$
where $|e_{nm}\rangle =|e_{n}\rangle ⊗|e_{m}\rangle$, with
(7)$$e_{nm}\left(x,y\right)=e^{-ik_{0}\alpha_{n}x}e^{-ik_{0}\beta_{m}y}.$$
Taking into account Equation (Equation 6), the vector |f〉 of Equation (Equation 5) may be expressed in terms of the Fourier basis functions as:(8)$$|f\rangle =A^{kl}\Phi_{kl}^{nm}|e_{nm}\rangle .$$
From a numerical point of view, only a finite number of $\left(2M_{x}+1\right)\times \left(2M_{y}+1\right)$, ($\left(M_{x},M_{y}\right)\in N^{2}$) Fourier coefficients is kept by selecting a finite sequence of $\alpha_{n}$ and $\beta_{n}$ in the expression Equation (Equation 7):(9)$$\left\{\begin{array}{c} \alpha_{n}=\alpha_{0}+n\lambda /D_{x},n\in \left[-M_{x}:M_{x}\right]\\ \;\beta_{m}=\beta_{0}+m\lambda /D_{y},m\in \left[-M_{y}:M_{y}\right] \end{array}\right.$$
where $D_{x}$ (resp. $D_{y}$) is the period along the (O,x) (resp. (O,y)) axis and λ is the operating wavelength. The parameters $\alpha_{0}$ and $\beta_{0}$ are linked to the incident wave. The choice of the truncation orders, $M_{x}$ and $M_{y}$, strongly depends on the values of $D_{x}$ and $D_{y}$. A large-scale problem involves large values of $D_{x}$ and/or $D_{y}$, yielding large truncation orders $M_{x}$ and $M_{y}$ as well as prohibitive computation time and memory storage capacity. This is one of the main limitations of the Fourier modal method when a large-scale problem issue is targeted such as the one addressed when designing some classes of photonics devices as metasurfaces. In order to accurately design highly efficient metasurfaces, near field (evanescent modes) are required to account for complex interactions among its sub-wavelength-scale constituents (meta-atoms). These evanescent modes, which are high-frequency harmonic modes, generally are highly sensitive to the truncation orders $M_{x}$ and $M_{y}$. The larger the size of the computational domain, the larger the values of the truncation orders that should be taken into account in order to describe the rapid oscillations of the evanescent modes. However, increasing $M_{x}$ and $M_{y}$ yields high computational cost and memory storage capacity. One of the trivial ways to compute high harmonic modes without increasing the truncation orders $M_{x}$ and $M_{y}$ is to reduce the size of the computational domain. This issue is addressed in the next subsection.

### 2.3. Domain Decomposition Spectral Method: Direct Transform

Here, we aim to find an approximation $|\tilde{f}\rangle$ of |f〉 of Equation (Equation 8) on a domain $\tilde{\Omega }\subset \Omega$, of width $d_{x}$, $d_{y}$ with respect to (O,x) resp. (O,y), assuming that a set of eigenvalues and eigenvectors are known for a boundary value problem stated on $\tilde{\Omega }$:(10)$$\tilde{L}|{\tilde{\Phi }}_{pq}\rangle =-{\tilde{\gamma }}^{2}|{\tilde{\Phi }}_{pq}\rangle .$$
We set $|\tilde{f}\rangle$ as
(11)$$|\tilde{f}\rangle ={\tilde{A}}^{pq}{\tilde{\Phi }}_{pq}^{\eta \mu }|{\tilde{e}}_{\eta \mu }\rangle ,$$
where
(12)$${\tilde{e}}_{\eta \mu }\left(x,y\right)=e^{-ik_{0}{\tilde{\alpha }}_{\eta }x}e^{-ik_{0}{\tilde{\beta }}_{\mu }y}$$
with ${\tilde{\alpha }}_{\left[\eta \right]}=\alpha_{0}+\left[-\eta_{x}:\eta_{x}\right]\lambda /d_{x}$ and ${\tilde{\beta }}_{\left[\mu \right]}=\beta_{0}+\left[-\mu_{y}:\mu_{y}\right]\lambda /d_{y}$. The weighted coefficients ${\tilde{A}}^{pq}$ (Equation (Equation 11)) are computed using a Galerkin-weighted-residual method. It can be easily shown that:(13)$$\left[{\tilde{A}}^{pq}\right]=\left[P_{kl}^{vw}\right]\left[A^{kl}\right],$$
where P is equal to:(14)$$\left[P_{kl}^{vw}\right]={\left[{\tilde{\Phi }}_{pq}^{\eta \mu }\right]}^{-1}{\left[\langle {\tilde{e}}_{vw}^{†},{\tilde{e}}_{\eta \mu }\rangle \right]}^{-1}\left[\langle {\tilde{e}}_{vw}^{†},e_{nm}\rangle \right]\left[\Phi_{kl}^{nm}\right].$$
Here, P clearly appears as the projector of the space spanned by ${\left(|\Phi_{kl}\rangle =\Phi_{kl}^{nm}|e_{nm}\rangle \right)}_{kl}$ onto the subspace spanned by ${\left(|{\tilde{\Phi }}_{pq}\rangle ={\tilde{\Phi }}_{pq}^{\eta \mu }|{\tilde{e}}_{\eta \mu }\rangle \right)}_{pq}$. In the current case of orthonormal Fourier basis functions, the set of $|{\tilde{e}}_{nm}\rangle$ satisfies $\langle {\tilde{e}}_{vw}^{†},{\tilde{e}}_{nm}\rangle =\delta_{v-n}\delta_{w-m}$. Therefore, Equation (Equation 14) yields
(15)$$\left[P_{kl}^{vw}\right]={\left[{\tilde{\Phi }}_{pq}^{\eta \mu }\right]}^{-1}\left[\langle {\tilde{e}}_{vw}^{†},e_{kl}\rangle \right]\left[\Phi_{kl}^{nm}\right].$$
Here, we define the inner product of two periodic or pseudo-periodic functions $g_{n}$ and $h_{m}$ defined on an interval $\left[x_{a},x_{b}\right]$ as:(16)$$\langle g_{n}^{†},h_{m}\rangle =\frac{1}{x_{b}-x_{a}}\int_{x_{a}}^{x_{b}}g_{n}^{†}\left(x\right)h_{m}\left(x\right)dx,$$
where $g_{n}^{†}$ denotes the complex conjugate of $g_{n}$. The inner product of Equation (Equation 15) may then be expressed as
(17)$$\left\{\begin{array}{c} {\langle {\tilde{e}}_{v}^{†},e_{k}\rangle }_{x\in [a,b]}=e^{-i\left(x_{b}+x_{a}\right)\tau_{x}}\mathrm{sinc}\left[\left(x_{b}-x_{a}\right)\tau_{x}\right]\\ {\langle {\tilde{e}}_{w}^{†},e_{l}\rangle }_{y\in [a,b]}=e^{-i\left(y_{b}+y_{a}\right)\tau_{y}}\mathrm{sinc}\left[\left(y_{b}-y_{a}\right)\tau_{y}\right] \end{array}\right..$$
where $\tau_{x}=\left(\frac{v}{d_{x}}-\frac{k}{D_{x}}\right)$, $\tau_{y}=\left(\frac{w}{d_{y}}-\frac{l}{D_{y}}\right)$. The definition of the inner product can be extended to the case of a two-dimensional (2D) problem by using the Kronecker product:(18)$$\langle {\tilde{g}}_{vw}^{†},h_{kl}\rangle =\langle {\tilde{g}}_{v}^{†},h_{k}\rangle \cdot \langle {\tilde{g}}_{w}^{†},h_{l}\rangle .$$
Let us consider now that the bounded plane Ω is squared into several elementary sub-cells $\Omega ={⋃}_{i,j}{\tilde{\Omega }}_{ij}$, with ${\tilde{\Omega }}_{ij}=\left[x_{i},x_{i+1}\right]\times \left[y_{j},y_{j+1}\right]$. Let us assume the known spectral coefficients ${\tilde{A}}^{pq}\left(i,j\right)$, related to each sub-domain ${\tilde{\Omega }}_{ij}$ such as any restriction of the expanded function ${\tilde{f}}_{ij}$ describing any component of the electromagnetic field on ${\tilde{\Omega }}_{ij}$, is written as:(19)$$|{\tilde{f}}_{ij}\rangle ={\tilde{A}}^{pq}\left(ij\right){\tilde{\Phi }}_{pq}^{\eta \mu }|{\tilde{e}}_{\eta \mu }\rangle .$$
The proposed DDSM allows us to obtain access to the near-field (i.e., electric and magnetic field) distribution of the device under study using Equation (Equation 19). However, in modeling an electromagnetic device, far-field information may be required in order to capture an accurate picture of the structure’s performance. Within the framework of the spectral method presented here, the problem we face consists of reconstructing the spectral coefficients of the electromagnetic field on the whole domain Ω, knowing the coefficients on each sub-domain ${\tilde{\Omega }}_{ij}$ obtained from the direct spectral decomposition transformation. This inverse transform issue will be detailed in the next subsection.

### 2.4. Domain Decomposition Spectral Method: Inverse Transform

The process we denoted as inverse transform consists of computing the global coefficient $A^{kl}$. These spectral coefficients are required to describe any electromagnetic field component function *f* in the whole space as:(20)$$f\left(x,y,z\right)=\sum_{kl}A^{kl}e^{-ik_{0}\gamma_{kl}z}\sum_{n,m}\Phi_{kl}^{nn}e_{nm}\left(x,y\right).$$
Using once again the Galerking-weighted-residual method, one can demonstrate that the set of $A^{kl}$ is obtained from ${\tilde{A}}^{pq}\left(ij\right)$ throughout the following matrix relation:(21)$$\left[A^{kl}\right]=\left[\begin{array}{cccc} Q(1,1) & Q(1,2) & \ldots & Q(N,N) \end{array}\right]\left[\begin{array}{c} {\tilde{A}}^{pq}\left(1,1\right)\\ {\tilde{A}}^{pq}\left(1,2\right)\\ \vdots \\ {\tilde{A}}^{pq}\left(N,N\right) \end{array}\right]$$
with
(22)$$\left[Q_{vw}^{kl}\left(i,j\right)\right]={\left[\Phi_{kl}^{nm}\right]}^{-1}\left[\langle e_{kl}^{†},{\tilde{e}}_{vw}\rangle \left(i,j\right)\right]\left[{\tilde{\Phi }}_{pq}^{\eta \mu }\right].$$
We summarize in Figure 2 the main steps of DDSM when it is applied to compute near and far fields of a photonics device in the framework of modal methods. For a better understanding, we also report in Figure 3 the flowchart and parallelization structure of the employed method [24]. As a first step, the computational domain Ω is squared into sub-domains ${\tilde{\Omega }}_{ij}$. Secondly, using the direct transform, the incident field is projected on the eigenvectors defined on each ${\tilde{\Omega }}_{ij}$ in the incident medium. Forward simulations can then be independently performed for each sub-domain, yielding an approximation of the near field on each ${\tilde{\Omega }}_{ij}$. Afterwards, the set of simulated near fields on each sub-cell is stitched together to reproduce the electromagnetic field on the whole surface. Finally, the inverse transform can be used to obtain the spectral coefficients of the field related to the whole domain if needed.

Having introduced the mathematical formalism of the proposed approach, we are now ready to go further into some applications.

## 3. Applications

### 3.1. DDSM Applied to a Dipole Field Expansion in the Framework of Modal Method

The purpose of the present section is merely to show what the DDSM looks like when applied to a simple case. In a Cartesian coordinates system (O,x,y,z), we are interested in the footprint of the field radiated by an electric dipole located at the origin O(0,0,0) in a given plane $z=-z_{s}$, as shown in Figure 4a. This plane will be the 2D computation domain Ω. This kind of simulation may be useful while performing the so-called adjoint simulation in an optimization problem of metalenses. In the case of the forward simulation, the device under optimization is excited by a planewave, while the adjoint simulation required the use of a dipole source located at the desired foci Figure 4b. In the framework of modal expansion, any component of the electromagnetic field of this dipole source can be fully described in the whole space by a set of known expansion coefficients $A^{pq}$, $B^{pq}$ and a set of eigenvectors $|\Phi_{pq}^{nm}\rangle$ as:(23)$$|f\left(z\right)\rangle =\left[A^{pq}\Phi_{pq}^{nm}e^{ik_{0}\gamma_{pq}z}+B^{pq}\Phi_{pq}^{nm}e^{-ik_{0}\gamma_{pq}z}\right]|e_{nm}\rangle .$$
For our illustration, only the first part of the above expression describing the downward wave is kept:(24)$$|f\left(z\right)\rangle =A^{pq}\Phi_{pq}^{nm}e^{ik_{0}\gamma_{pq}z}|e_{nm}\rangle .$$
The set of eigenvectors $|\Phi_{pq}^{nm}\rangle$ and their associated eigenvalues $\gamma_{pq}$ can be computed thanks to Equation (Equation 1). Practical details for the numerical computation of the weighted coefficients $\left[A^{pq}\right]$ and $\left[B^{pq}\right]$ can be found in [25]. We report in Figure 5a,b, the phase and the real part, respectively, of $E_{x}\left(x,y,z_{s}=-20\lambda \right)$, in vacuum $\forall \left(x,y\right)\in \left[-0.5D_{x},0.5D_{x}\right]\times \left[-0.5D_{y},0.5D_{y}\right]$. The truncation orders $M_{x}$ and $M_{y}$ are set to $M_{x}=M_{y}=31$, $D_{x}=D_{y}=26.67\lambda$.

A typical well-defined wavefront of a dipole source can be distinguished. These results will serve as a baseline and will be compared to those obtained with the proposed domain decomposition method. First, the plane $z=-z_{s}$ is squared into several elementary sub-cells $\Omega ={⋃}_{i,j}{\tilde{\Omega }}_{ij}$, with ${\tilde{\Omega }}_{ij}=\left[x_{i},x_{i+1}\right]\times \left[y_{j},y_{j+1}\right]$. Then, the eigenvalue equation Equation (Equation 10), obtained from Maxwell’s equations, equipped with PMLs (perfectly matched coordinates), are numerically and independently solved in each sub-cell. From a geometrical point of view, in a homogenous medium, the eigenvalue Equation (Equation 10) only depends on the size of the computation area. Consequently, for a set of sub-cells ${\tilde{\Omega }}_{ij}$ of the same size, only one numerical simulation based on Equation (Equation 10) is performed to obtain the sequence of eigenvectors $|{\tilde{\Phi }}_{pq}\rangle$. The truncation orders are set to $m_{x}=m_{y}=10$. The sequence of the spectral coefficients $\left[{\tilde{A}}^{pq}\left(i,j\right)\right]$ in each sub-cell ${\tilde{\Omega }}_{ij}=\left[x_{i},x_{i+1}\right]\times \left[y_{j},y_{j+1}\right]$ is then obtained from the weighted coefficients $\left[A^{pq}\right]$ using Equation (Equation 13). The restriction of any component of the electromagnetic field $|f_{i,j}\left(z\right)\rangle$ on a sub-cell ${\tilde{\Omega }}_{ij}$ is then written as:(25)$$|{\tilde{f}}_{\left(i,j\right)}\left(z\right)\rangle ={\tilde{A}}^{pq}\left(i,j\right){\tilde{\Phi }}_{pq}^{nm}e^{ik_{0}{\tilde{\gamma }}_{pq}z}|{\tilde{e}}_{nm}\rangle .$$
The set of the simulated fields on each sub-cell is then stitched together (geographically, without any post-processing) to reproduce the electromagnetic field on the whole surface Ω.

Figure 6a,b represent the real parts and the phase, respectively, computed using Equation (Equation 25) by dividing the computational domain $\left[-0.5D_{x},0.5D_{x}\right]\times \left[-0.5D_{y},0.5D_{y}\right]$ into 3×3 sub-domains. In Figure 7a,b, this number is extended to 5×5 sub-domains. In all these figures, where no post-processing treatment is applied, the field distribution in the sub-cells areas and the effect of the PMLs areas are clearly shown. Comparing these results with the baseline results of Figure 5a,b, one can also remark that out of the PMLs areas (ratio PMLs-area-wide/$\Omega_{ij}$-wide<<1/10), the footprints of the dipole source are well-described on each sub-cell ${\tilde{\Omega }}_{ij}$ by the sequence of computed local spectral coefficients $\left[{\tilde{A}}^{pq}\left(i,j\right)\right]$. The method then shows its ability to represent the footprint of a given electromagnetic field component. This leads to the next step where we are ready to deploy the proposed concept to compute the electromagnetic field diffracted by a large-scale device. Three potential devices are investigated: a dielectric Fresnel Plate Zone (FPZ), a metalens made of dielectric cylindrical waveguides (nanorodes) with variables cross-sections and a Pachatarnam–Berry phase dielectric metalens.

### 3.2. Modal Method Applied to a Large Binary Fresnel Plate Zone (Binary FPZ)

In this section, the first proposed case will be presented. The task consists of a forward simulation of both the near and far-field components of a 2D dielectric binary FPZ with a height *h* under a linearly polarized incident plane wave, as shown in Figure 8a. The FPZ consists of a set of concentric dielectric rings, known as Fresnel zones, operating as a dielectric lens using diffraction to focus light into a given focal point. The considered dielectric material has a refractive index of 1.99 deposited on a ${\mathrm{SiO}}_{2}$ substrate (refractive index = 1.45). Its height is set to h=400 nm for an operating wavelength of λ=532 nm. To obtain a constructive interference at the focus point, the radius of the Fresnel zones should satisfy the equation:(26)$$R_{n}=k_{0}\sqrt{n\lambda f+{\left(\frac{n\lambda }{2}\right)}^{2}},$$
where *f* is the lens’ focal length, λ is the operating wavelength and $k_{0}=2\pi /\lambda$ is the wavenumber in the focusing medium. The aperiodic Fourier modal method (AFMM) [26,27,28], a full-wave analysis based on the Fourier modal method equipped with PMLs (perfectly matched layers) [29,30,31,32,33], is used to compute the electromagnetic field components. It is well known that in the case of the FMM or AFMM, the total computation time grows as $O(N^{6})$ (bi-periodic problem) with respect to the total truncation order *N* required for all the field components (i.e, the four field components used for the continuity equation). On the other hand, memory consumption time grows as $O(N^{4})$. Therefore, scaling the computation $\left[-0.5D_{x},0.5D_{x}\right]\times \left[-0.5D_{y},0.5D_{y}\right]$ surface into some sub-sections will certainly provide advantages to scaling both the computational time and the required memory capacities as $\left(2m_{x}+1\right)\left(2m_{y}+1\right)/\left(2M_{x}+1\right)\left(2M_{y}+1\right)$. In our illustration, the lens’ focal length is set to 70λ, and the number of FPZ rings is 15, yielding a 100λ-width device. To apply the DDSM to the proposed geometry, the 100λ×100λ×400 nm structure is divided into 5×5×400 nm sub-cells of $d_{x}\times d_{y}\times h=\;$6.7277λ×6.7277λ×400 nm sub-voxels. Each sub-voxel is treated independently, with very low truncation numbers $m_{x}$ and $m_{y}$, thereby enabling the simulation to be more feasible in terms of memory and time consumption. In the current case $\left(2m_{x}+1\right)\times \left(2m_{y}+1\right)=\left(2\times 12+1\right)\times \left(2\times 12+1\right)$, Fourier harmonics are enough to describe the electromagnetic field behavior on each sub-domain. On the contrary, the electromagnetic field expansion considering the whole domain Ω involves at least $\left(2M_{x}+1\right)\times \left(2M_{y}+1\right)=\left(2\times 62+1\right)\times \left(2\times 62+1\right)$ Fourier harmonics yielding a simulation that is extremely consuming in both time and memory. The insets in Figure 8c,d display the top-view layout of the FPZ lens and the (x,y)-plane phase distribution of the transmitted electric field on the emerging surface of the lens, respectively. Regarding Figure 8d, one can remark the full [−π,π] phase coverage characterizing a binary phase zone plate. Making good use of the inverse transform post-processing, we can obtain access to field distribution in the whole 3D space. In particular, we compute and display in Figure 8e the transmitted field in the 3D space. Regarding this result, one can remark that the incident electric field intensity at the foci considerably increases up to 3500 times. As reported in Table 1, the computational times for the simulation of the near field (i.e., emerged field on the FPZ surface) using the Fourier modal method 2D-AFMM implemented with $m_{x}=m_{y}$ Fourier basis functions (so that the size of the eigenvalues equations matrix is $2\left(2m_{x}+1\right)\times 2\left(2m_{x}+1\right)$) are 35.07 s for $m_{x}=m_{y}=9$, 57.29 s for $m_{x}=m_{y}=10$, 90.77 s for $m_{x}=m_{y}=11$, and 152.67 s for $m_{x}=m_{y}=12$ ($M_{x}=M_{y}=62$). These simulations are performed on a classical personal computer DELL PRECISION 3640, with an intel CORE i9 (3.10 Ghz) processor. Note that since the $\left[-0.5D_{x},0.5D_{x}\right]\times \left[-0.5D_{y},0.5D_{y}\right]$ surface is strongly inter-meshed, $2^{12}\times 2^{12}$ nodes have been used both for the highly pixelated profile of the FPZ lens surface and also for the computation nodes of the near field.

### 3.3. Modal Method and Large Metalens Consisted of Set of Different Cross-Sections Nanorods

The considered second example is a dielectric metalens consisting of a set of subwave- length-scale dielectric Si waveguides with a refractive index of 3.6082, which is deposited on a substrate with a refractive index of 2.4626 in Figure 8b. At a given point (x,y) on the metalens, the radius of each waveguide is chosen so that the spatial distribution of the phase profile θ(x,y) follows the following equation:(27)$$\theta \left(x,y\right)=k_{0}\left[f-\sqrt{x^{2}+y^{2}+f^{2}}\right].$$
The metalens height is set to h=5.57 μm and the operating wavelength is λ=7 μm. The lens is designed to focus on a linearly polarized incident plane wave at a focus length of f=10λ.

We perform forward simulation on a Ω=12.8571λ×12.8571λ-width device. The results are presented in Figure 9. The device surface is squared into 3×3 sub-domains $\Omega_{ij}$, as shown in Figure 9a. The truncation orders are set to $m_{x}=m_{y}=17$, which is equivalent to $M_{x}=M_{y}=52$ on Ω. A [−π,π] coverage is clearly distinguished in Figure 9b, where the phase distribution of the emerged electric field on the top face of the metalens is presented. Applying the inverse transform allows one to obtain the spectral coefficients of the global structure. The emerged near and far fields can then be computed and the focus action of the metalens can be visualized, as shown in Figure 9c.

In Figure 10a, and for the same device, we increase the number of the sub-domains from 3×3 up to 5×5. This allows one to decrease the truncation orders down to $m_{x}=m_{y}=10$ ($M_{x}=M_{y}=52$). As shown in Figure 10b,c, results related to this new partition are still accurate.

### 3.4. Large-Scale Pachatarnam–Berry Phase-Based (PB) Metalens Simulation

In the third example, we are interested in a Pachatarnam–Berry phase metalens. It consists of an arrangement of rotated dielectric meta-atoms having a predefined shape and a refractive index of 1.99 deposited on a SiO${}_{2}$ substrate (refractive index = 1.45). In our example, a rectangular shape is considered. The dimensions of the rectangular meta-atoms are (length=300 nm, width=105 nm) and the metalens height is h=400 nm. These parameters are chosen such that the PB phase-based metasurface converts and focuses a fraction of the left-hand circularly polarized (LCP) incident light injected from the substrate into right-hand circular polarization (RCP). To obtain the desired in plane (x,y) phase distribution for focusing, each meta-atom located at position (x,y) in the plane is rotated at an angle θ(x,y) with respect to (O,x) axis such as
(28)$$\theta \left(x,y\right)=\frac{1}{2}k_{0}\left[f-\sqrt{x^{2}+y^{2}+f^{2}}\right].$$

The operating wavelength is set to λ=0.532µm. The simulation was run on the same personal computer as the previous case. Results are presented in Figure 11. The computation domain Ω=23.30λ×23.30λ is squared into 3×3 sub-domains leading to $d_{x}=d_{y}=7.76\lambda$-width sub-cells. The phase distribution of the co-polarization $E_{x}^{RCP}\left(x,y\right)$, $E_{y}^{RCP}\left(x,y\right)$ components of the emerged RCP electric field at the top face of the metalens, obtained using the direct and the inverse transform, are presented in Figure 11c,d. A [−π,π] coverage is clearly distinguished. Applying the inverse transform yields in Figure 11b, an accurate reproduction of the transmitted power of the metalens.

Finally, we increase the PB-based metalens up to 30.6λ×30.6λ width. One can easily admit that increasing the lens size will lead to a more efficient device. The spatial distribution of the transmitted power and phases of the electric field of the cross-polarization are shown in Figure 12. As expected, both the power energy flow and the resolution of the designed lens are increased compared to the previous case of a 23.30λ width metalens (Figure 11). However, increasing the lens’ geometrical features requires increasing the number of sub-cells in order to keep a low value of the truncation numbers $m_{x}$ and $m_{y}$. The device is divided into 5×5 sub-cells of 6.16λ×6.16λ, where the truncation orders are set to $m_{x}=m_{y}=10$.

## 4. Conclusions and Outlook

To summarize, we introduce an approach, namely a domain decomposition spectral method DDSM, based on spectral methods, which can be optimally and simply integrated in a parallel computing scheme to provide a solution for large-size systems. On one hand, the power of a spectral method relies on the use of a few coefficients to describe both near and far-electromagnetic fields in a three-dimensional space. On the other hand, modal methods provide a better representation of physical phenomena involving modal interactions such as resonance phenomena through dielectric and metallic periodic and aperiodic metasurfaces. While many previous works, mainly based on spatial domain methods such as finite difference time and frequency domain methods, finite elements methods, etc., address large-scale problems efficiently, despite some previous developments, the simulation of large-scale aperiodic devices remains a challenging problem for full spectral modal methods. We expect that our work will help to tackle this issue. In order to show the efficiency of the proposed work, we provided numerous examples, such as an electric dipole simulation and some monochromatic metalenses illuminated under both circular and linear polarized incident plane waves. In fact, the proposed method is based on two processes: the direct and inverse transforms, which allow one to compute, efficiently, both near and far-field spectrums. The DDSM direct transform consists of decomposing the spectral coefficients of the total field onto some spectral coefficients related to the sub-domains. Meanwhile, the inverse transform consists of reconstructing the spectral coefficients of the total field starting from the coefficients of decomposition of each field obtained from the direct spectral decomposition transformation.

Our domain decomposition spectral method provides a promising pathway for handling large aperiodic, but it is still limited, and further enhancements are needed to extend this method to a very, very large-scale problem. Indeed, the process of decomposition of the incident field requires the resolution of a unique eigenvalue equation stated on a very-very wide domain. However, in an electromagnetism problem, the incident field is not unknown to the problem. This field is generally well known, and its modal characteristic can be efficiently and reliably computed based on interpolating methods.

Therefore, we plan to calculate the modal characteristics of incident fields based on neural network models. This strategy could ultimately and reliably increase the performance of the method in the case of very, very large-scale problems.

## Figures and Tables

**Figure 1 sensors-22-08131-f001:**
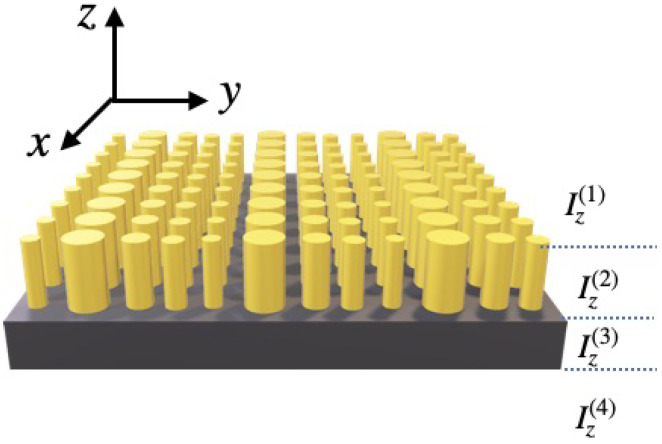
Sketch of a 2D grating. In the framework of the modal method, the 2D structure is divided into 4 layers $I_{z}^{\left(1\right)}$, $I_{z}^{\left(2\right)}$, $I_{z}^{\left(3\right)}$ and $I_{z}^{\left(4\right)}$ with respect to the propagation direction that is assumed to be (O,z).

**Figure 2 sensors-22-08131-f002:**
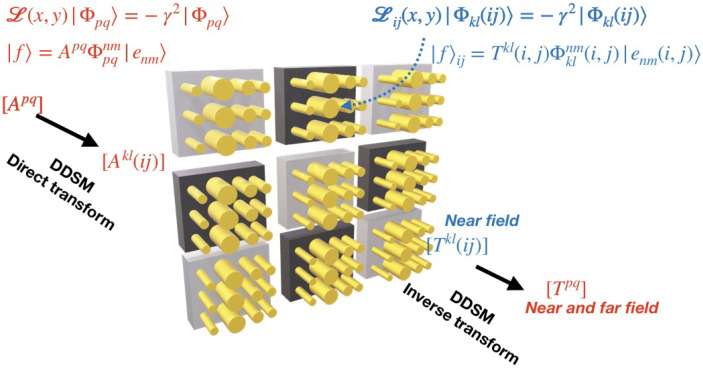
Sketch of the direct and inverse transform of DDSM applied to the near and far fields simulation of a photonics device.

**Figure 3 sensors-22-08131-f003:**
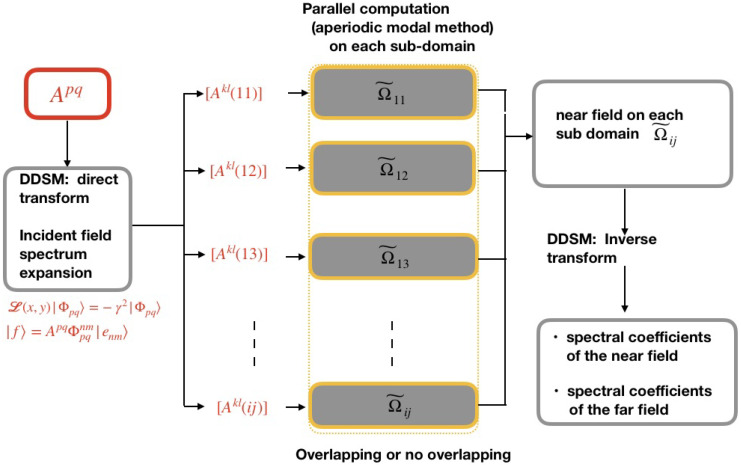
Flowchart and parallelization structure of the direct and inverse transform of DDSM applied to the near and far fields simulation of a photonics device.

**Figure 4 sensors-22-08131-f004:**
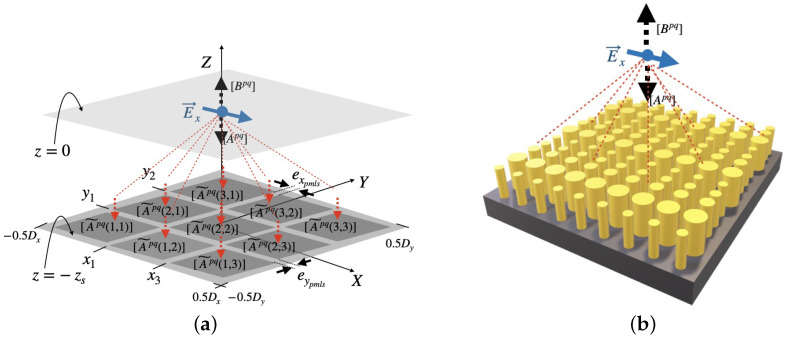
Sketch of a dipole field decomposition. The fundamental strategy of the DDSM to compute the footprint of a dipole field on an arbitrary large-area surface consists in dividing the surface into elementary sub-cells. Any component of the dipole field can then be locally simulated, on each sub-cell. Here, the surface $z=-z_{s}$ is subdivided into 3×3 sub-cells. (**a**) Dipole field decomposition and recomposition. (**b**) Sketch of simulation of metasurface under a dipole illumination.

**Figure 5 sensors-22-08131-f005:**
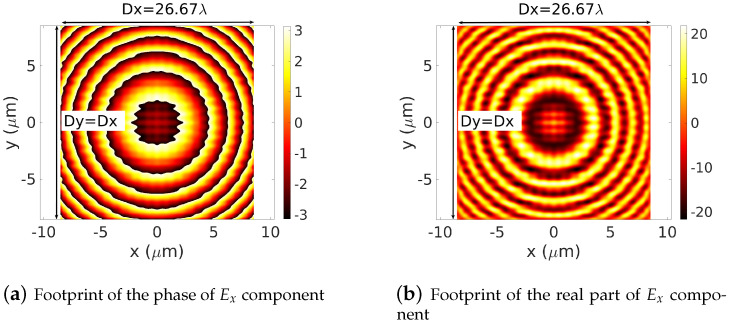
Footprints of the phase (**a**) and the real part (**b**) of an $E_{x}$ electric dipole on the plane z=−20λ. The numerical computation is performed on the whole domain $\left[-0.5D_{x},0.5D_{x}\right]\times \left[-0.5D_{y},0.5D_{y}\right]$. The operating wavelength: λ=0.64 μm, $D_{x}=D_{y}=26.67\;\lambda$, truncation orders $M_{x}=M_{y}=31$.

**Figure 6 sensors-22-08131-f006:**
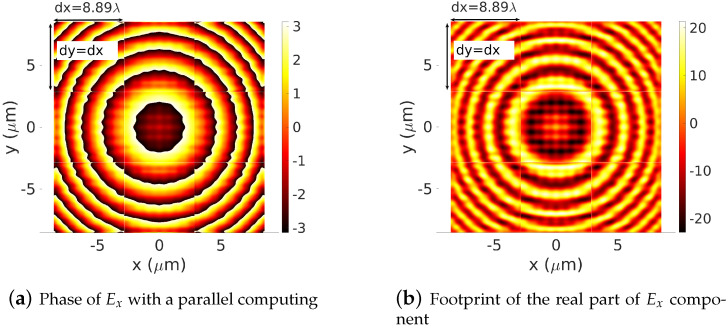
Footprints of the phase (**a**) and the real part (**b**) of an $E_{x}$ electric dipole in the plane z=−20 λ. The whole domain $\left[-0.5D_{x},0.5D_{x}\right]\times \left[-0.5D_{y},0.5D_{y}\right]$ is divided into a 3×3 sub-cell $\Omega_{ij}$. Based on the proposed parallel strategy, the numerical computation is performed on each sub-cell $\Omega_{ij}$. The operating wavelength: λ=0.64 μm, $d_{x}=d_{y}=8.89\;\lambda$, truncation orders $m_{x}=m_{y}=10$.

**Figure 7 sensors-22-08131-f007:**
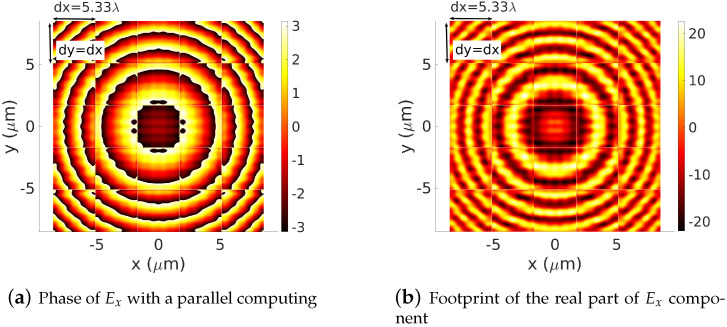
Footprints of the phase (**a**) and the real part (**b**) of an $E_{x}$ electric dipole in the plane z=−20λ. The whole domain $\left[-0.5D_{x},0.5D_{x}\right]\times \left[-0.5D_{y},0.5D_{y}\right]$ is divided onto a 5×5 sub-cell $\Omega_{ij}$. The operating wavelength: λ=0.64 μm, $d_{x}=d_{y}=5.33\;\lambda$, truncation orders $m_{x}=m_{y}=6$.

**Figure 8 sensors-22-08131-f008:**
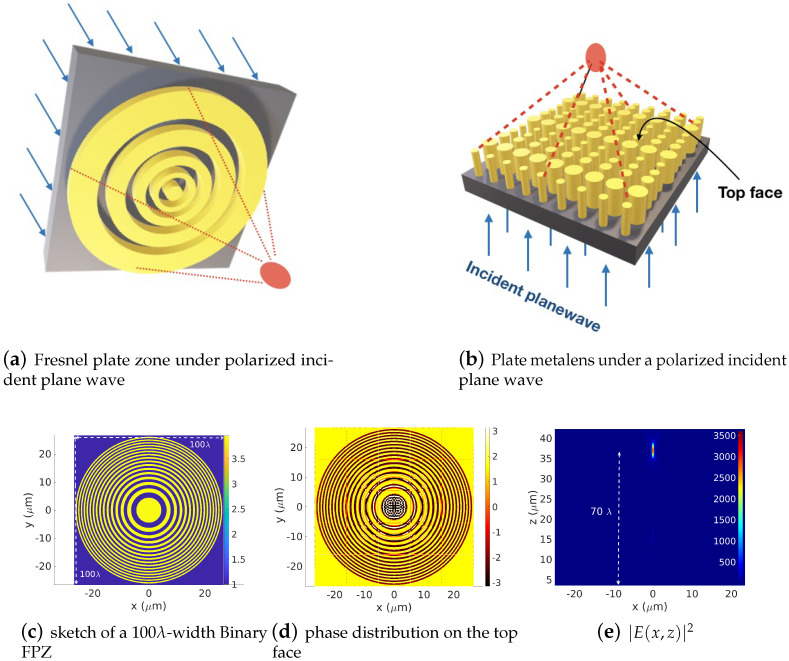
A forward simulation of a 100λ-width FPZ using the DDSM. The computation domain is subdivided into 5×5 sub-domains $\Omega_{ij}$. (**c**) presents the sketch of the diffractive Fresnel lens, (**d**) shows the phase distribution on the top face of the lens and (**e**) displays the transmitted field. Numerical parameters: λ=0.532 μm, $z_{s}=70\lambda$, n=31 zones (15 rings), $D_{x}=D_{y}=100\lambda \;$h=400 nm, refractive index of the lens material 1.99, refractive index of the substrate 1.45, $m_{x}=m_{y}=12$, ($M_{x}=M_{y}=62$).

**Figure 9 sensors-22-08131-f009:**
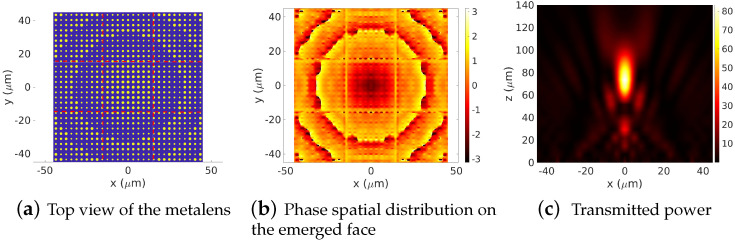
Forward simulation of a metalens consisting of sub-wavelength dielectric nanorodes with different cross-section. Power distributions of the focusing spot in the (x,y=0,z) plane, λ=7 μm, $z_{s}=10\lambda$, domain Ω=12.8571λ×12.8571λ=3×3 sub-cell $\Omega_{ij}$, h=5.57 μm, materials: dielectrics Si 3.6082 deposited on material with refractive index (2.4626), $m_{x}=m_{y}=17$, $M_{x}=M_{y}=52$.

**Figure 10 sensors-22-08131-f010:**
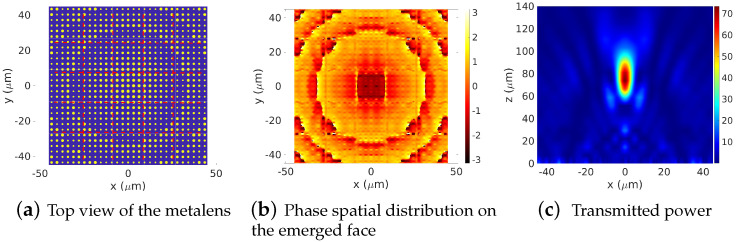
Forward simulation of a metalens consisting of sub-wavelength dielectric nanorodes with different cross-section. Power distributions of the focusing spot in the (x,y=0,z) plane, λ=7 μm, $z_{s}=10\lambda$, domain Ω=12.8571λ×12.8571λ=5×5 sub-cell $\Omega_{ij}$, h=5.57 μm, materials: dielectrics Si ν=3.6082 deposited on material with refractive index 2.4626, $m_{x}=m_{y}=10$, $M_{x}=M_{y}=52$.

**Figure 11 sensors-22-08131-f011:**
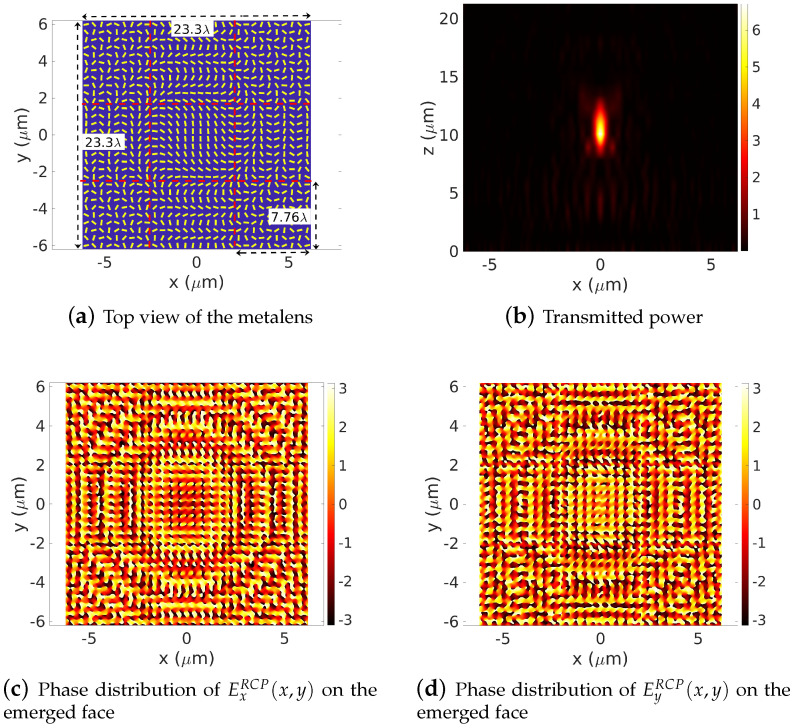
Forward simulation of a Ω=23.3λ×23.3λ-width Pachatarnam–Berry phase-based metalens. The lens is designed to have a focus length of f=20λ. The number of sub-cells and truncation orders are set to 3×3 and $m_{x}=m_{y}=12$ ($M_{x}=M_{y}=37$), respectively. (**b**) displays the power distribution in the (x,y=0,z) plane. The phase distribution of the co-polarization $E_{x}^{RCP}\left(x,y\right)$, $E_{y}^{RCP}\left(x,y\right)$ components of the emerged RCP electric field at the top face of the metalens, obtained using the direct and the inverse transform, are presented in (**c**,**d**). A [−π,π] coverage is clearly distinguished. Numerical parameters: λ=0.532 μm, focus length $z_{s}=20\lambda$, $D_{x}=D_{y}=23.30\lambda$, h=400 nm.

**Figure 12 sensors-22-08131-f012:**
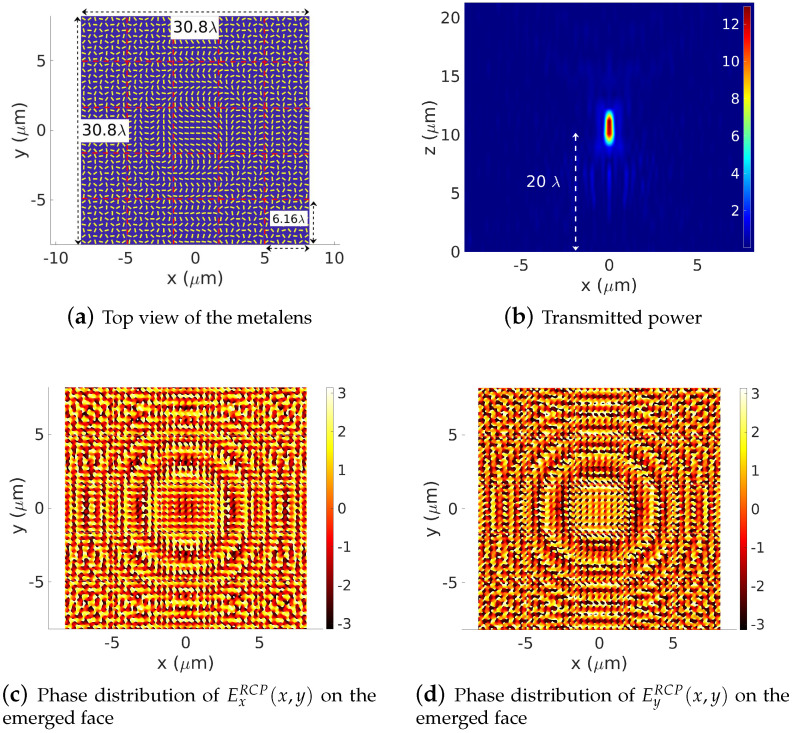
Forward simulation of a Ω=30.8λ×30.8λ-width Pachatarnam–Berry phase-based metalens. The lens is designed to have a focus length of f=20λ. Power distributions of the focusing spot in the (x,y=0,z) plane. Phase distribution of $E_{x}\left(x,y\right)$ component of the cross-polarization (RCP) on the top of the PB-based metalens. The number sub-cells and truncation orders are set to 5×5 and $m_{x}=m_{y}=10$, respectively. Numerical parameters: λ=0.532 μm, focus length $z_{s}=20\lambda$, *h* = 400 nm.

**Table 1 sensors-22-08131-t001:** Panel of the computation times for the simulation of the FPZ near field on each sub-cell using the aperiodic Fourier modal method (AFMM). Simulation is performed on a classical computer DELL PRECISION 3640 with a processor intel CORE i9 (3.10 Ghz). The whole $\left[-0.5D_{x},0.5D_{x}\right]\times \left[-0.5D_{y},0.5D_{y}\right]$ surface is strongly inter-meshed $2^{12}\times 2^{12}$ nodes have been used both for the highly pixelated profile the FPZ lens surface and also for the computation nodes of the near field.

$m_{x}=m_{y}$	9	10	11	12
computation times (s)	35.07	57.29	90.77	152.67

## Data Availability

Further Computational Details about the Method’s Implementation along with a Version of the Code. Available online: photonicsnum.com (accessed on 26 September 2022).

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
