# Peer review of "Domain Decomposition Spectral Method Applied to Modal Method: Direct and Inverse Spectral Transforms"

_sensors, 2022, doi:10.3390/s22218131_

Round 1

Reviewer 1 Report

The authors present a work entitled "Domain decomposition spectral method applied to the modal method: Direct and Inverse spectral transforms," where they show an approximate method for solving Maxwell's equations for large systems.

The article is interesting for readers, has solid mathematical development, and presents a series of proof examples of the method.

However, given the computational efficiency that the authors propose, I recommend adding a section or appendix with the computational details of the implementation of the method. For example, flowchart, parallelization structure, or code repository.

I hope that the authors agree to make the computational tool available to readers, given the interest in solving this type of system.

I believe that the article can be published in the magazine sensors.

Reviewer 2 Report

In this work, the authors introduced a domain decomposition spectral method DDSM, which can be optimally and simply integrated in a parallel computing scheme to provide a solution for large-size systems. Good results are obtained. The power of a spectral method relies in the use of a few coefficients to describe both near and far-electromagnetic fields in a dimensional space. This method has great potential applications in dielectric and metallic periodic and aperiodic metasurfaces. I think it can be accepted to be pulished.
